

# Accurate image-based identification of macroinvertebrate specimens using deep learning—How much training data is needed?

Toke T. Høye[1,2], Mads Dyrmann[3], Christian Kjær[1], Johnny Nielsen[1], Marianne Bruus[1], Cecilie L. Mielec[1], Maria S. Vesterdal[1], Kim Bjerge[3], Sigurd A. Madsen[4], Mads R. Jeppesen[4] and Claus Melvad[2,4]

[1] Department of Ecoscience, Aarhus University, Aarhus, Denmark
[2] Arctic Research Centre, Aarhus University, Aarhus, Denmark
[3] Department of Electrical and Computer Engineering, Aarhus University, Aarhus, Denmark
[4] Department of Mechanical and Production Engineering, Aarhus University, Aarhus, Denmark

Corresponding author
Toke T. Høye, tth@ecos.au.dk

## ABSTRACT

Image-based methods for species identification offer cost-efficient solutions for biomonitoring. This is particularly relevant for invertebrate studies, where bulk samples often represent insurmountable workloads for sorting, identifying, and counting individual specimens. On the other hand, image-based classification using deep learning tools have strict requirements for the amount of training data, which is often a limiting factor. Here, we examine how classification accuracy increases with the amount of training data using the BIODISCOVER imaging system constructed for image-based classification and biomass estimation of invertebrate specimens. We use a balanced dataset of 60 specimens of each of 16 taxa of freshwater macroinvertebrates to systematically quantify how classification performance of a convolutional neural network (CNN) increases for individual taxa and the overall community as the number of specimens used for training is increased. We show a striking 99.2% classification accuracy when the CNN (EfficientNet-B6) is trained on 50 specimens of each taxon, and also how the lower classification accuracy of models trained on less data is particularly evident for morphologically similar species placed within the same taxonomic order. Even with as little as 15 specimens used for training, classification accuracy reached 97%. Our results add to a recent body of literature showing the huge potential of image-based methods and deep learning for specimen-based research, and furthermore offers a perspective to future automatized approaches for deriving ecological data from bulk arthropod samples.

# INTRODUCTION

Image-based approaches to the study of arthropod specimens are rapidly gaining attention (*Høye et al., 2021*; *Lürig et al., 2021*). New computer vision tools can provide

morphological information such as body size or more specific features such as wing size or body length for pinned specimens (*Wilson et al., 2022*). Images of specimens can also be used for taxonomic identification by training classification models, although the accuracy depends on the methods and the image quality (*Ärje et al., 2020b; Raitoharju & Meissner, 2019; Raitoharju et al., 2018*). Images are typically generated by photographing pinned specimens (*Hansen et al., 2020; Ströbel et al., 2018; Wilson et al., 2022*), by imaging individually manipulated specimens (*Ärje et al., 2020a; Wührl et al., 2022*), or by imaging mixed samples directly in trays (*Blair et al., 2020; Schneider et al., 2022*), on sticky traps (*Geissmann et al., 2022; Gerovichev et al., 2021*), or even under natural conditions (*Bjerge, Mann & Høye, 2022; Bjerge et al., 2021; Mungee & Athreya, 2020; Pegoraro et al., 2020*). With the rise of convolutional neural networks (CNNs), classification performance has reached remarkable levels when these models are trained on high quality image data (*Ärje et al., 2020a; Ärje et al., 2020b*), but a recurrent conclusion is that accuracy depends on the amount of training data.

Species identification from images is an example of fine-grained classification in computer vision terminology, and state-of-the-art deep learning models capable of accurately performing such tasks require large amounts of training data for regularization of the many parameters of CNNs (*Luo et al., 2018*). Generally, it appears that larger training data sets produce the highest classification performance. Such data sets should be balanced (*i.e.,* where the sample size of specimens are evenly distributed across taxa) to avoid bias unrelated to the visual characteristics of each species. However, class imbalance where some taxa are rare and others are common (*Johnson & Khoshgoftaar, 2019*) is found in most biological samples, and it is, therefore, challenging to produce balanced training data sets with large sample sizes for rare species (*Ärje et al., 2020a*). For this reason, it is important to get a better understanding of how classification accuracy relates to the number of specimens of each species in the training of classification models through rigorous tests, and whether other factors can further influence classification results.

More closely related species are likely to be harder to separate for classification models, as well as higher taxonomic levels more easily reach higher levels of classification accuracy. Similarly, smaller species may be harder to identify correctly than larger species, simply because they are represented by fewer pixels in the images. Indeed, a recent study of 361 species of carabid beetles demonstrated that specimen size was a very important predictor of classification accuracy (*Hansen et al., 2020*). The same study also showed how classification accuracy increased considerably when species-level classifications were only evaluated to the level of genus rather that to the species level. However, their dataset had considerable class imbalance and did not test the effect of sample size on classification performance.

Here, we describe such a test for a balanced set of 16 taxa of freshwater macroinvertebrates of relevance to water quality assessments. The dataset consists of two Coleoptera, two Ephemeroptera, two Malacostraca, five Plecoptera, one Sphaeriidae, three Trichoptera, and one planarian (Tricladida). The objectives of this study were to (1) evaluate the influence of reduced sample size on the accuracy of image-based classification of 16 freshwater invertebrate taxa, (2) determine if weighted methods of assigning taxonomic identity increases classification accuracy compared to non-weighted methods, and (3)

determine if image-based classification accuracy varies among taxa and is related to body size. To achieve these objectives, we evaluate a series of deep learning models trained on balanced sets of training data differing only in the number of specimens of each taxon used in the model training. All models were evaluated on the same training data. Since each specimen was imaged from multiple angles, we were also able to evaluate whether integrating all images of a specimen in the classification through majority vote increased classification accuracy compared to simple classification of each image separately.

## MATERIALS AND METHODS

### Field and lab work

The specimens used in this study were collected in March 2021. We focused on several taxa of relevance to the assessment of ecological status of streams. All specimens were identified by morphology by co-author JN, who is an expert in stream insect identification. All specimens were collected in Denmark. Coleoptera, Plecoptera and Trichoptera were collected in Brook Brandstrup, and Ephemeroptera were collected in Stream Mattrup. These two running waters are part of the River Gudenå watercourse system. All Malacostraca and Sphaeriida were collected in Brook Holmehave, which is part of the River Odense watercourse system. The identification by morphology was supported by *Dall (1990)*, and the level of taxonomic identification is presented in Table 1. For the taxa represented only by juvenile specimens, we included specimens representing as broad variation in life stages as possible in order to obtain a robust categorization.

### Imaging specimens with BIODISCOVER

We used the BIODISCOVER (BIOlogical specimens Described, Identified, Sorted, Counted and Observed using Vision-Enabled Robotics) machine for imaging specimens and image analysis (*Ärje et al., 2020a*). In short, this system consists of an aluminium case with two Basler ACA1920-155UC cameras and VS Technology LD75 lenses. The cameras are placed perpendicular to each other in two opposite corners of the case. Each camera records at a frame rate of 50 frames per second with an exposure time of 2,000 µs. In the third corner, a high-power LED light (ODSX30-WHI Prox Light) is aimed at the opposite and final corner with a three cm tall and 1 cm ×1 cm wide cuvette made of optical glass and filled with ethanol. The aluminium case has a lid and is rubber-sealed to minimize extraneous light, shadows, and other disturbances. The BIODISCOVER is controlled by a computer with an integrated software. The software detects the specimen through a blob detection algorithm. The blob is essentially the silhouette of the specimen in a particular image. During the imaging process, selected geometric features of the blob such as its area, perimeter, and maximum diameter are calculated for each image and stored in a text file. Mean area across all images of a specimen was used to characterize its body size. The blob is also used to crop the images to 496 pixels width (defined by the width of the cuvette) and 496 pixels height while keeping the specimen at the centre of the image with regards to the height of the blob. If a specimen exceeds the height of 496 pixels, the resulting images are cropped with a greater height. The images are stored as PNG files. Further details are described in *Ärje et al. (2020a)*.

**Table 1   The taxa included in the study.** For each taxon, the table provides a species name if specimens could be identified to the species level. The table also gives the number of species of each genus known from Denmark, the number of specimens of each taxon included in this study and the mean number of images recorded with the BIODISCOVER system for each specimen.

| Order | Family | Genus | Species | Species in genus in DK | Specimens | Mean number of images per specimen |
|---|---|---|---|---|---|---|
| Coleoptera | Elmidae | Elmis | *Elmis aenea* | 1 | 61 | 162 |
| | | Limnius | *Limnius volckmari* | 2 | 64 | 67 |
| Ephemeroptera | Baetidae | Baetis | *Baetis rhodani* | 5 | 64 | 132 |
| | Heptageniidae | Heptagenia | | 5 | 102 | 61 |
| Malacostraca | Asellidae | Asellus | *Asellus aquaticus* | 1 | 63 | 67 |
| | Gammaridae | Gammarus | *Gammarus pulex* | 2 | 66 | 43 |
| Plecoptera | Nemouridae | Amphinemura | | 2 | 60 | 360 |
| | Taeniopterygidae | Brachyptera | *Brachyptera risi* | 2 | 65 | 166 |
| | Perlodidae | Isoperla | | 2 | 73 | 161 |
| | Leuctridae | Leuctra | | 4 | 83 | 202 |
| | Nemouridae | Nemoura | | 4 | 69 | 257 |
| Sphaeriida | Sphaeriidae | Sphaerium | | 3 | 70 | 16 |
| Trichoptera | Rhyacophilidae | Rhyacophila | | 2 | 61 | 102 |
| | Sericostomatidae | Sericostoma | *Sericostoma personatum* | 1 | 81 | 40 |
| | Goeridae | Silo | | 2 | 75 | 200 |
| Tricladida | Dugesiidae | Dugesia | | 2 | 63 | 89 |

Each specimen is dropped in the ethanol-filled cuvette and photographed multiple times as it sinks and passes through the field of view of the cameras. Since the framerate is constant, the number of images decreases with the sinking speed. The sinking speed varies with the density and shape of the specimen. All images belonging to the same specimen are grouped so that they are only included in either training, validation, or testing, and thus no information leakage occurs between the datasets. The number of specimens and the average number of images per specimen for each species are presented in Table 1.

## Model training

We trained the model EfficientNet-B6 end-to-end. EfficientNet is a CNN architecture that is characterized by high accuracy for the given number of parameters. EfficientNet has previously demonstrated state-of-the-art classification accuracies on the CIFAR-100 dataset, while having fewer trainable parameters than other architectures in the same accuracy range (*Tan & Le, 2019*). CIFAR-100 is a dataset of small images containing 100 different classes (*Krizhevsky, 2009*). Images were scaled to 224×224 pixels, as it made it possible to use pre-trained network weights. It also allowed us to fit a different number of classes than in the original paper without making substantial changes to the network architecture. Images were also augmented randomly with rotation, flipping of image axes, and shear to increase the variance in the training data. Shearing is a shift of one or both image axes, which causes straight lines to be preserved while angles change, *i.e.*: $\begin{pmatrix} x' \\ y' \end{pmatrix} = \begin{pmatrix} 1 & \lambda_x \\ \lambda_y & 1 \end{pmatrix} \begin{pmatrix} x \\ y \end{pmatrix}$, where $\lambda_x$ and $\lambda_y$ are the vertical and horizontal displacement factors

(*Gonzalez & Woods, 2017*). Prior to each training, the network is initialized with weights pre-trained on ImageNet (*Deng et al., 2009*).

We selected 10% of the total number of specimens from the training set for validation. During the model training, we monitored the validation loss (*i.e.,* the change in classification error on the validation dataset for each training epoch) to use the network weights of the training iteration, where the loss reached its minimum. At this stage, the training was stopped, and the network weights were extracted and used to test model performance on the independent test data set. As stop criterion, we used classification accuracy for all images in the validation set, and we used cross-entropy as loss function.

We trained the CNN classification model with different training data sets to evaluate the classification accuracy across the taxa involved in the experiment and evaluate how the number of specimens influences the accuracy. In each model, we used the same number of specimens for training across all taxa. We trained models using five, 10, 15, 20, 25, 30, 35, 40, 45, and 50 specimens of each taxon. We note that the number of images used in the training of models was far greater, as each specimen was represented by up to several hundred images.

## Model evaluation

All CNN models were evaluated on their ability to make correct classifications on a test set. The test set consisted of images from ten specimens of each taxon not used during the training and validation process. We made sure that for all models, the test images were always the same. In addition to training the network with varying numbers of training specimens, we performed 10-fold cross-validation of the full dataset to estimate variation in the results following resampling.

We evaluated model performance using three metrics: Accuracy, Precision, and Recall. *Accuracy* indicates the proportion of correct classifications of all samples. *Precision* for class $x$ is the proportion of correct predictions of all the samples that were classified as $x$, i.e. $TP_x/(TP_x + FP_x)$. A high precision for class $x$ thus indicates that few samples are misclassified as class $x$, but it does not take into account samples that were missed. *Recall* for class $x$ is the proportion of samples from class $x$, which are classified as $x$, i.e., $TP_x/(TP_x + FN_x)$. A high recall indicates that most samples from class $x$ are classified correctly. We calculated precision and recall for each taxon separately and evaluated how the mean precision and mean recall changed in response to the sample size of the training data.

Each specimen is represented by a series of images, and each image can be classified individually. We can, therefore, evaluate the performance of the models by evaluating the per-image accuracy or by majority-voting among all images for each specimen (*Johnson & Khoshgoftaar, 2019*). With voting, the specimen is assigned to the taxon that has received the most votes among the images of the specimen. Since the images are taken from two orthogonal angles, and since the insects often rotate on their way through the cuvette, there may be images taken from angles where it is easier to recognize the taxon than others. Thus, there may be misclassifications of some of the images, which are compensated for in voting, if the majority of the classifications of the specimen is correct.

We carried out voting in two ways that differ in the weight assigned to each vote. In majority voting, each image of a specimen is assigned equal weight, and the specimen is assigned to the taxon that gets the highest number of votes. In max scoring sum, the votes are weighted by scores of the neural network. The output of the neural network is a vector with an unbounded score value for each of the 16 taxa. The greater the number of image features that are activated for a given taxon, the greater the score for that taxon. When voting with weighted scores, the output vectors for all images belonging to a specimen are summed, and the taxon with the highest total sum of scores is assigned to the individual. Summarizing these score values has the advantage of accounting for the uncertainty of the network. Images that the network finds hard to recognize will typically have low scores for all taxa, and taxa that are easily confused will all have weights in the same range.

## RESULTS

We imaged at least 60 specimens of each of the 16 taxa of freshwater arthropods (Fig. 1). Due to differences in sinking speed, the mean number of images recorded per specimen varied among taxa, with fewest images for Sphaerium (16) and most images for Amphinemura (360). In total, we recorded 148,228 images across 1120 specimens (Table 1). Taxa also varied in body size, and some taxa such as *Sericostoma personatum* were represented by multiple age classes and accordingly varied considerably in body size (Fig. 2).

There was a marked increase in precision when using majority voting or when using max score sum compared to unweighted precision (Fig. 3). The biggest advantage of using weighting was seen in the results with few training samples, where there was a difference of four to seven percentage points between weighted and unweighted precision. This difference became smaller as the number of training images increases, which means that the models made fewer mistakes, and the advantage of weighing images from each individual was reduced (Fig. 3). Since the majority vote gave slightly better results than the max scores sum, we only present subsequent results using majority vote. Cross-validation with 10 splits showed good consistency among different test sets. Mean accuracy when using majority voting was 99.2% (range 97.2% –100.0%) (Fig. S1).

With an increasing number of specimens used for training, the classification accuracy gradually increased to 99% when trained on 50 specimens of each taxon (Fig. 4). We found a high classification accuracy of the CNN model even when trained on very few specimens. Indeed, the model trained on only 15 specimens of each taxon exhibited a classification accuracy of 97% (Fig. 3).

Taxa in orders with only one or two representatives in the dataset were classified correctly even with smaller samples of training data. For other taxa, precision generally increased with an increased number of training individuals. (Fig. 5). With the small number of taxa in this data set and the generally very high accuracy, we found no evidence that taxa with smaller body sizes were harder to classify than taxa with larger body sizes. We do note though that our dataset included taxa with highly variable body sizes (Fig. 2). As training data increased, only taxa in Plecoptera were mixed up, and this was the only order represented by five taxa in the dataset. The species with the highest error rates are Silo and Brachyptera with 95.8% and 96.6% average recall, respectively (Fig. S2).

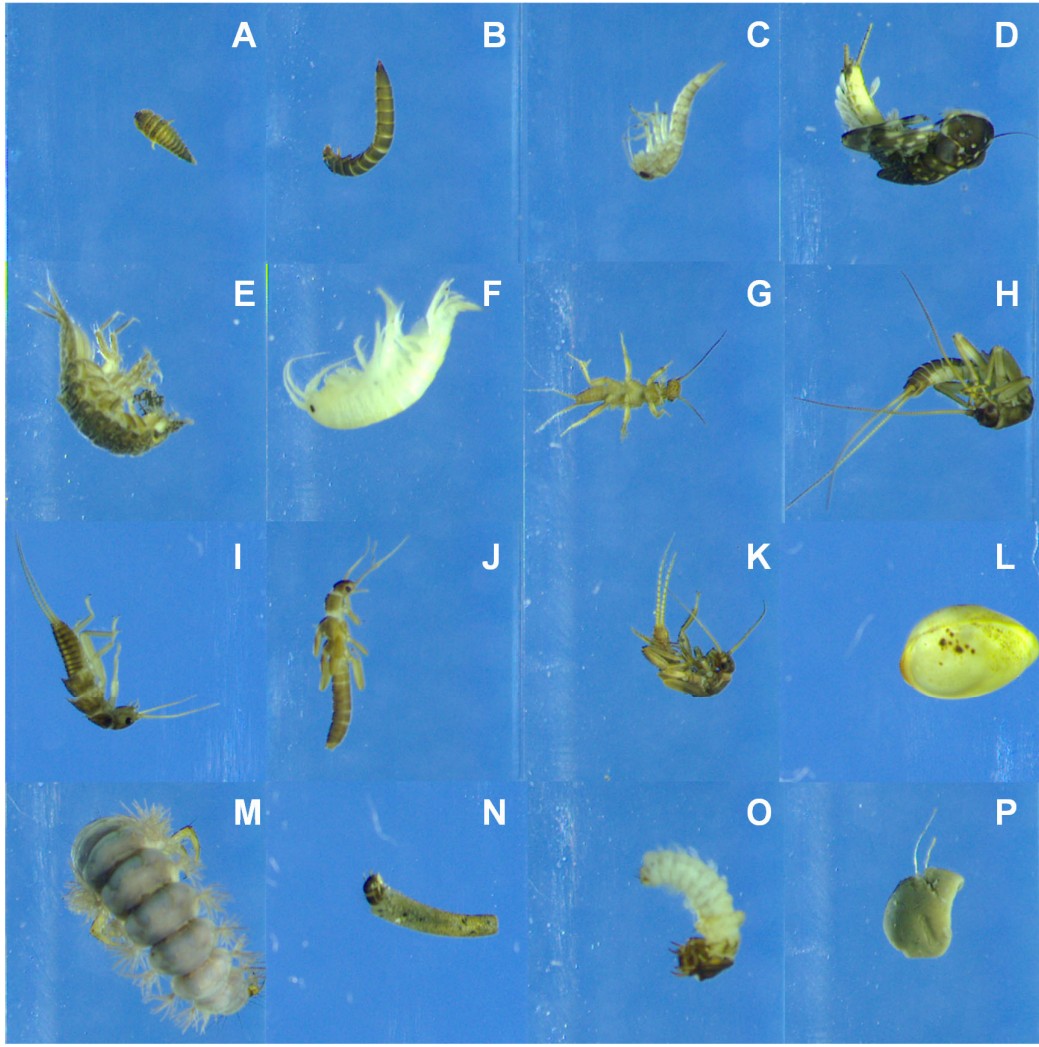

**Figure 1** Example photos recorded by the BIODISCOVER of each of the taxa included in the study in the order in which they appear in Table 1. (A) *Elmis aene*, (B) *Limnius volckmari*, (C) *Baetis rhodani*, (D) Heptagenia, (E) *Asellus aquaticus*, (F) *Gammarus pulex*, (G) Amphinemura, (H) *Brachyptera risi*, (I) Isoperla, (J) Leuctra, (K) Nemoura, (L) Sphaerium, (M) Rhyacophila, (N) *Sericostoma personatum*, (O) Silo, (P) Dugesia.

## DISCUSSION

We have made a first assessment of how the classification accuracy is affected by sample size for image-based identification of arthropods. We achieved a remarkable 99.2% classification accuracy with 50 specimens of each taxon in the training data. Even with only 15 specimens of each taxon in the training data, we achieved 97% accuracy. We have previously shown 98% accuracy on a dataset of terrestrial arthropods with lower sample sizes (*Ärje et al., 2020a*), and the results of this study extend the very high classification accuracy obtained with the BIODISCOVER system to freshwater arthropods. We find that by integrating the classification across multiple images of the same specimen taken from
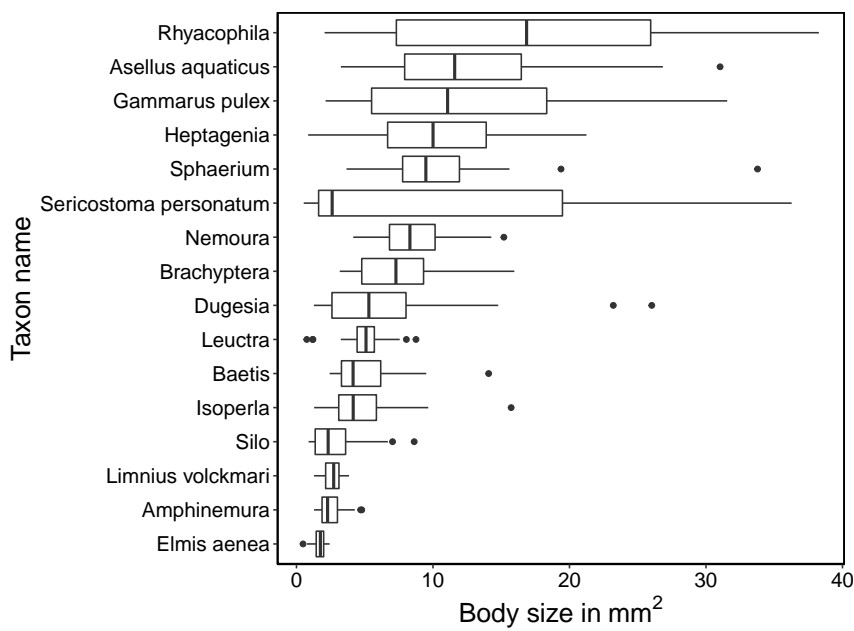

**Figure 2** **Boxplots of the variation in mean size (in mm²) of specimens.** Size is estimated by the mean area in pixels covered by the specimen in each image taken by the BIODISCOVER during its passage through the ethanol-filled cuvette. Size in pixels is translated into mm² based on the estimate that 10 mm width equals 496 pixels resulting in an approximate pixel size of $4.06 \times 10^{-4}$ mm².

different angles using majority vote, we improve classification accuracy considerably. This suggests that important details are captured by imaging specimens from multiple angles with systems such as the BIODISCOVER (*Ärje et al., 2020a*) or by generating actual 3D models of specimens (*Ströbel et al., 2018*). In comparison, identification systems based on nadir images of bulk insect samples may have faster processing times, but are likely to miss important morphological details (*Blair et al., 2020*; *Schneider et al., 2022*). It could be interesting to examine if images from specific angles are more important, and also to use attention models to investigate which characteristics the model uses to recognize and distinguish different species, and how these compare to human experts (*Wührl et al., 2022*).

Deep learning is finding many applications in ecology and evolution, but an important bottleneck is the requirement for accurate and abundant training data (*Besson et al., 2022*; *Christin, Hervet & Lecomte, 2019*; *Lamba et al., 2019*). Our results suggest that the high image quality and the imaging from multiple angles provided by the BIODISCOVER system reduces the requirement for large sample sizes in the training data. This appears to be true even for taxa with small or highly variable body sizes. It is, therefore, feasible to collect sufficient samples even of rare taxa for training powerful classification models, which could be able to separate among a larger number of taxa. The relationship between classification accuracy and sample size is likely to change, if we increase the number of taxa. In particular, morphologically similar classes may increase the demand for training data. In the future, more demanding tests on a larger number of taxa can evaluate how error rates among species may depend on the number of closely related species (*e.g.*, in the

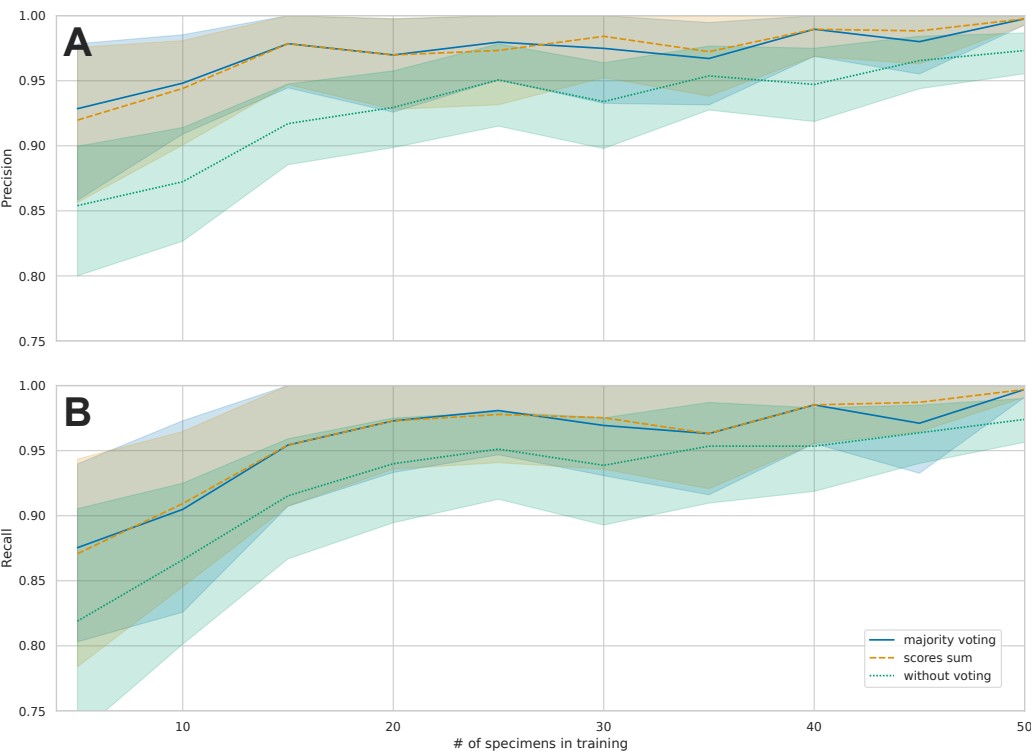

**Figure 3** **Comparison of the mean precision and recall across species for the three different principles of classification.** Individual images, majority vote and scores sum. (A) shows the average precisions across the 16 taxa and the 95% confidence intervals. When increasing the number of training samples, the precision also increases. B: This figure shows the average recall across the 16 taxa and the 95% confidence intervals. When increasing the number of training samples, the recall also increases.

same genus). The image data from this study is open access and can be used in such future comparison (*Høye et al., 2022*).

Due to the relatively small number of taxa in this study, we were unable to fully evaluate how body size influenced precision, but there was some indication that precision was lower for smaller species. A previous study identified body size as important for classification performance, but the measure of body size was somewhat indirect, as specimens were not segmented from the background, and image size of an automatically generated rectangular bounding box was used as a crude proxy for body size (*Hansen et al., 2020*). In contrast, the BIODISCOVER system generates an accurate proxy for body size automatically during the imaging process through blob detection (*Ärje et al., 2020a*). As reference collections accumulate, it will be straightforward to evaluate the importance of body size in combination with other factors for the classification performance of deep learning models.

The diversity of stream macroinvertebrates is widely used as indices of ecological quality (*Birk & Hering, 2006*) or effects of organic pollution (*Friberg et al., 2010*), pesticides (*Beketov et al., 2009*), and urban development (*Gadd et al., 2020*). The quality of such assessments is found to be sensitive to the sample size and whether all species are identified and counted (*Ligeiro et al., 2020*). Sampling must be rigorous to be able to

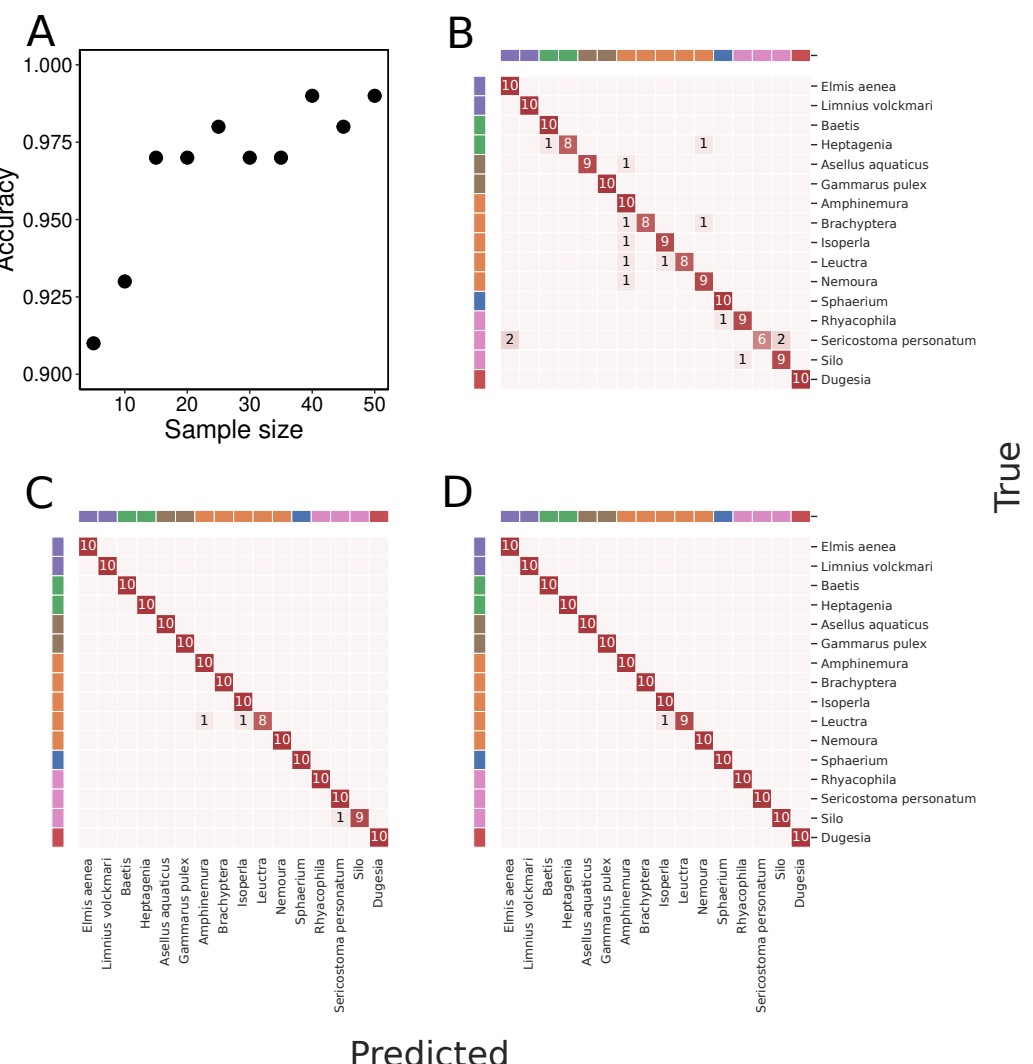

**Figure 4 Overall classification accuracy at different amounts of training data and example confusion matrices.** (A) presents overall classification accuracy as a function of varying the number of specimens of each taxa used for training. The classification accuracy is based on the majority vote principle. With 50 specimens of each taxon used for training, all test specimens except one were correctly assigned to each taxon and classification accuracy was thus 99%. Classification results presented as confusion matrices using (B) 5, (C) 25, and (D) 50 specimens of each taxon for training. The taxa are ordered as in Table 1, and taxa sharing the same colour in the band along the perimeter of the confusion matrices belong to the same order. In all three panels, the testing is done on images of the same 10 specimens of each taxon not used in the training process.

pick up ecological signals in the inherently variable fauna samples, but at the same time, sampling has to be cost-effective (*Ramos-Merchante & Prenda, 2017*; *Vlek, Sporka & Krno, 2006*). The promising results of the present study suggest that automated species identification, counting, size distribution, and biomass estimation may allow for a more rigorous assessment of the ecological status of streams in the future (*Ärje et al., 2020a*; *Ärje et al., 2020b*; *Raitoharju et al., 2018*). The BIODISCOVER system does not yet offer an
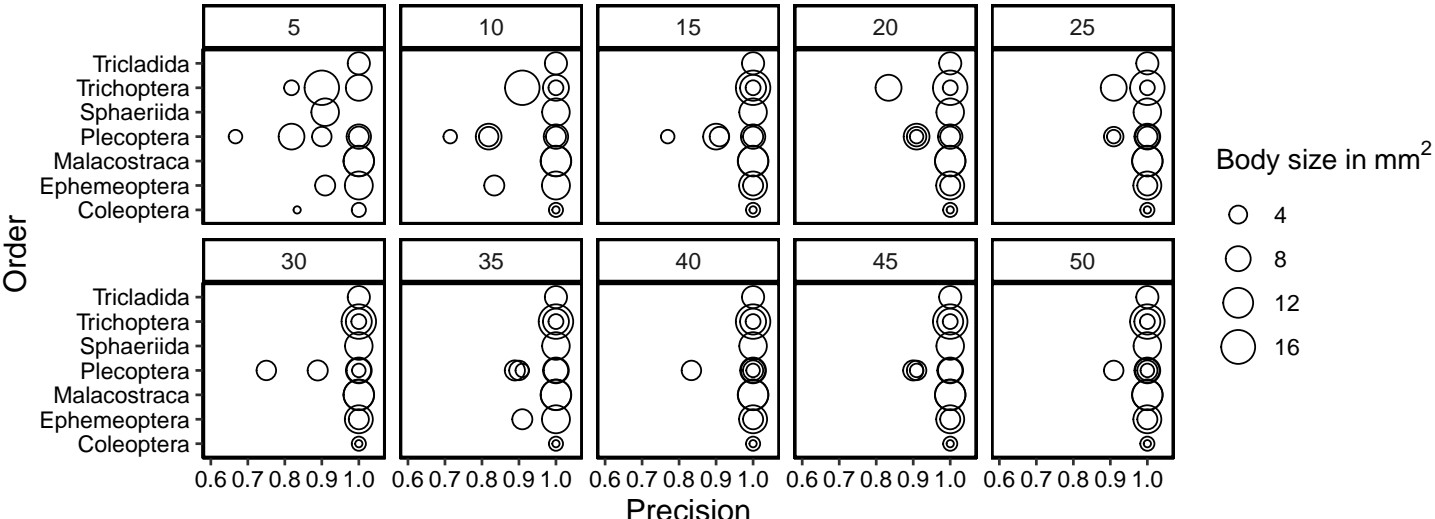

**Figure 5** **Variation in precision for taxa in each order included in the study.** Each panel shows the precision for a given number of training samples and the dot size is scaled according to the mean body size of the taxon.

automated solution for separation of debris and invertebrates, nor does it yet automate the feeding of specimens to the system. Although separating the sampled invertebrates from debris and sorting them into main groups is a time-consuming part of a typical sample treatment in Danish stream monitoring for regulatory purposes, species identification accounts for roughly half of the sample handling time. As such, there are immediate benefits to the system, but also a potential for even further benefits to come. In the future, a growing image library could allow for fast identification of bulk samples. When automated identification is refined and an autosampler installed, sample handling time will be reduced considerably. This would allow for *e.g.*, collection of more samples within the same economic frame, leading to greater spatio-temporal resolution of community data, which would make assessments of ecological status from bulk samples more precise. It might also facilitate a better understanding of the individual and synergistic effects of multiple environmental stressors to stream ecosystems (*Beermann et al., 2021*). For monitoring purposes, it is of paramount importance that identifications are comparable across observations from different sites and time points. Automated procedures will increase the reproducibility of the identification by reducing variation in identification from different observers. Such differences arise due to individual variation in taxonomic skills and through potential differences in sorting procedures among laboratories. In addition, the easier access to identification of stream invertebrates at species level may facilitate scientific projects studying these organisms, because the demand for taxonomic expertise will be lowered. Such benefits go beyond freshwater macroinvertebrates and can help generate comparable data across aquatic and terrestrial invertebrate samples including insects.

## ACKNOWLEDGEMENTS

Kristian Meissner, Finnish Environment Institute is acknowledged for leading the DETECT project, which developed the prototype for BIODISCOVER.

### Funding

This work was supported by strategic funds from the Department of Ecoscience, Aarhus University. The funders had no role in study design, data collection and analysis, decision to publish, or preparation of the manuscript.

### Grant Disclosures

The following grant information was disclosed by the authors:
The Department of Ecoscience, Aarhus University.

### Competing Interests

The authors declare there are no competing interests.

### Author Contributions

- Toke T. Høye conceived and designed the experiments, analyzed the data, prepared figures and/or tables, authored or reviewed drafts of the article, and approved the final draft.
- Mads Dyrmann conceived and designed the experiments, performed the experiments, analyzed the data, prepared figures and/or tables, authored or reviewed drafts of the article, and approved the final draft.
- Christian Kjær conceived and designed the experiments, prepared figures and/or tables, authored or reviewed drafts of the article, and approved the final draft.
- Johnny Nielsen conceived and designed the experiments, performed the experiments, prepared figures and/or tables, and approved the final draft.
- Marianne Bruus conceived and designed the experiments, authored or reviewed drafts of the article, and approved the final draft.
- Cecilie L. Mielec performed the experiments, authored or reviewed drafts of the article, and approved the final draft.
- Maria S. Vesterdal conceived and designed the experiments, authored or reviewed drafts of the article, and approved the final draft.
- Kim Bjerge performed the experiments, authored or reviewed drafts of the article, and approved the final draft.
- Sigurd A. Madsen performed the experiments, authored or reviewed drafts of the article, improved the BIODISCOVER hardware and software to allow for efficient imaging, and approved the final draft.
- Mads R. Jeppesen performed the experiments, authored or reviewed drafts of the article, improved the BIODISCOVER hardware and software to allow for efficient imaging, and approved the final draft.

- Claus Melvad performed the experiments, authored or reviewed drafts of the article, improved the BIODISCOVER hardware and software to allow for efficient imaging, and approved the final draft.

## Data Availability

The data is available at Zenodo: Høye, Toke T., Dyrmann, Mads, Kjær, Christian, Nielsen, Johnny, Bruus, Marianne, Mielec, Cecilie L., Vesterdal, Maria S., Bjerge, Kim, Madsen, Sigurd A., Jeppesen, Mads R., & Melvad, Claus. (2022). BIODISCOVER image data on Danish freshwater macroinvertebrates. Zenodo. https://doi.org/10.5281/zenodo.6380934.

## Supplemental Information

Supplemental information for this article can be found online at http://dx.doi.org/10.7717/peerj.13837#supplemental-information.

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
