# Peer review of "Accurate image-based identification of macroinvertebrate specimens using deep learning—How much training data is needed?"

_PeerJ, doi:10.7717/peerj.13837_

## Round 0.1 · original submission · Minor Revisions

The reviewers both indicate that the manuscript is acceptable, but some revisions are required - in particular, I agree with Reviewer 2, who has requested that you include some discussion around the subject of the practical benefits of this approach in biomonitoring, including a critical evaluation of the practicality of the approach, and how it compares to other methods (e.g. microscopy, molecular methods etc).

·

Basic reporting

Clear and well written with only a few clarifications required or suggested:
> Line 136-137 - expand upon this line to explain why images were scaled from the 496x496 mentioned above to the 224x224 format, and what shear and flipping are for readers unitiiated in CNN training.
> Line 146 - It would feel simpler to say "Each model was then tested on images form the same ten specimens of each taxa not used during the training and validation process."
> Line 199-201 - I don't follow how confusion between these pairs is evident from Figure 4. There is some mislassification with just 5 specimens but with more speciemens the models appear to discriminate among these pairs. Figure S2 on the other hand does show precision, but is high for all taxa except Isoperla. The first part of S2 shows recall is lower for Brachyptera, S.personatum and Silo than some others, but again Nemoura is higher than Isoperla and its not clear from this where the confusion lies.
> Line 203 - "where" to "were"
> Line 218 - "angels" to "angles"
> The legend for Figure 4 should proably say the coloured bands around the perimeter indicate which taxa belong to the same orders

Experimental design

As a whole the article is well defined, but I think one area of ambiguity is in the distinction between training specimens and training images. By imaging a specimen many times and incorporating further flipping and shearing the training dataset is far greater than the number of specimens implies. I therefore think its worth being a little clearer that in the full-models the training dataset is composed of many hundreds of images.

Validity of the findings

The findings are very robust, remarkably so as the authors put it, but it is clear these results may not always be replicated under other circumstances with more diverse assemblages. Nonetheless the conclusions drawn are well supported.

Reviewer 2 ·

Basic reporting

1) The article is well written in general, but clarity may benefit from consistent edits and word choice throughout. Please disregard the following examples if the writing style is typical of the journal.
Excessive use of the word “the” throughout. Word count could be significantly reduced if removed. For example: “the” at the beginning of the sentence in line 193 could be removed to read as “Mean accuracy when using majority voting was 99.2% (range 97.2% –194 100.0%) (Figure S1).” Also change “is” to “was” throughout to reflect past versus present tense.

Consider replacing the word “specimen(s)” with “individual(s)” throughout.

2) Consider reorganizing by including additional subheadings. Currently, the topics of model training, validation, precision, accuracy, sensitivity, and performance are scattered and unorganized. Organizing these important topics into unique subheadings in the methods and results will greatly improve clarity for the reader. I believe the text is all there but clarity is lost in the current organization. Below is an example to consider. Adjust accordingly as I may have missed important details…..

MATERIALS AND METHODS

Field data collection
Lines 90-99

Imaging of specimens with BIODISCOVER
Lines 101-119

Model description
Briefly describe both EfficientNet-B6 and CNN and how they are connected. Currently there isn’t enough info for the reader. Is EfficientNet-B6 a process for training CNNs?

Model development
Describe the model training and validation process. Also, should the cross-validation to get at sensitivity be included here (lines 170-172) instead of among the model performance section? Also, reducing the training data and measuring correct classification could also be considered sensitivity. Clarify how are these are different?

Model performance
Describe model performance in general. Then the test data (lines 146-148). Then the specific measures used herein to evaluate performance. Keep the description of accuracy, precision, and recall separate. Maybe even assign additional subheadings. Include how results will be interpreted and what each means to clarify for the reader. Include things like What does it mean if recall decreases with smaller sample size? Etc.

RESULTS
Training data
General characteristics of the training data lines 179-184
Break out results in similar or same subheadings as above.

3) literature cited is appropriate although lines 218-222 should either be softened or removed as the authors do not directly evaluate the other methods.

4) tables and figures are appropriate although figure 5 includes body size which is mostly glossed over in the ms. Authors should shore up the importance of body size in the manuscript to justify including a dedicated figure in the ms or move the figure to SI. Call out body size as a subheading and organize. Fig 3 also includes body size but only 1 sentence is dedicated to it in the results.

Experimental design

1) Lines 78-86 describe the general purpose of the study but do not clearly spell out the objectives/research question(s). Clearly spelling out the objectives as objectives will greatly improve the quality of the ms and help guide the authors to better organize the ms for clarity (see above comments). For example: The objectives of this study were to 1) evaluate the influence of reduced sample size on the accuracy and precision of image-based classification of 16 freshwater invertebrate taxa, 2) determine if weighted methods of assigning taxonomic identity increases classification accuracy compared to non-weighted methods, and 3) determine if image-based classification varies among taxa. Adjust as necessary. Use objectives to guide and organize methods and results. Remove any text in the ms that doesn’t support the objectives.

2) Reorganization by adding subheadings as described above will help clarify and focus the methods.

Validity of the findings

1) Interpretation of the results align with the presented tables and figures.

2) The appeal of the ms will be greatly widened if the authors include a section or even a case study type scenario in the discussion detailing the application of image-based identification of macroinvertebrates for a typical stream bio-monitoring program setting. Walk the reader through the steps, discuss advantages, and limitations. Include a description of the type of field sampling methods where this method would be beneficial. Based on several decades of experience working with benthic samples, I don’t see the direct benefit for a monitoring program where most of the cost associated with stream invert samples is time in the lab separating invertebrates from debris so they can be identified and counted and not in the actual identification step - but I very well may be missing something. I believe this method has merit in other types of studies such as biodiversity or museum work but not necessarily for typical biomonitoring programs. Additional discussion of direct benefits to monitoring programs will help clarify the application of this approach in taxonomic identification.

Additional comments

no further comments.

---

## Round 0.2 · accepted · Accept

Your revisions have greatly improved the manuscript, and I am pleased to recommend publication of this interesting study.